# Reinforcement Learning Based Adaptive Walking Assistance Control of a Lower Limb Exoskeleton

**Yingying Wang, Yutian Shen**
Department of Electronic Engineering
The Chinese University of Hong Kong
Shatin, Hong Kong
{1155135738, 1155113996}@link.cuhk.edu.hk

## Abstract

Lower limb exoskeleton for people suffering from unilateral weakness or even hemiparesis has attracted considerable interest in recent years. For walking assistance in such scenarios, the exoskeleton is expected to help generate symmetrically coordinated motion with the unaffected side. However, control strategies remains a challenge as the exoskeleton system is an under-actuated system with some degree of freedoms remains passive and it includes human-machine interaction making it more complex than fully actuated robotic systems where classical PID controller can achieve satisfying performance. In this project, we propose a reinforcement learning based controller to provide walking assistance based on a modeled leader-follower system where the task can be reformulated as a motion trajectory tracking problem. The proposed control method is validated in simulation platform with data obtained from a healthy subject imitating hemiplegia patients with a pre-developed set of single-side lower limb exoskeleton. A video demonstration of this project is: RL based adaptive walking assistance control of a lower limb exoskeleton

## 1 Introduction

Lower limb exoskeleton, as an assistance device in strength augmentation [1], [2] and rehabilitation [3], [4], has gained considerable interest in the past few decades. Especially, stroke has become a global health problem that may cause weakness in upper and lower limbs, some researchers have focus on walking assistance or recovery help for people suffering from unilateral weakness or even hemiparesis.

Some researchers propose to use predefined trajectory of the healthy subjects and some use stored pattern of the unaffected leg of the subject [5] as reference to provide motion support. Some use different torque profiles according to phase of walking cycle, forward movement assistance during swing phase and joint stability augmentation during stance [6] for example. However, these methods are not robust to disturbances caused by human-exoskeleton system or external environment and the control performance is likely to be affected. Moreover, the predefined trajectory strategy suffers from fixed trajectory that can not be adapted to different subjects, while the torque control strategy relies heavily on accurate model which is difficult to obtain in real application scenarios. Facing these challenges, we propose to use reinforcement learning (RL) based control algorithm to provide adaptive walking assistance for subjects in different locomotion conditions.

In this project, a RL based controller is proposed to provide walking assistance that adapts to different subjects. Then a simulation platform with OpenAI Gym-compatible interface[7], which simulates the exoskeleton-human system, is built to validate the algorithm before implementing on the designed single-side lower-limb exoskeleton system to evaluate the performance.

The organization structure of the paper is as follows. Brief introduction to exoskeleton system is provided and the control strategies in related work is reviewed. The exoskeleton-human system is modelled as a leader-follower system, then the motivation and task is reformulated to optimal control problem in Section 2. The details of the RL based controller and the simulation platform is introduced in Section 3, then experimental results in the simulation with data partially collected from the exoskeleton-human system is presented in Section 4. Finally, conclusion and future work is summarized in Section 5

## 1.1 Related Work

Recently, artificial intelligence (AI) research has given rise to powerful techniques for RL. RL algorithms have been successfully applied to many complex aspects and proved even outperform human beings, such as playing chess. Many of the control-theoretic approaches and techniques for approximately solving optimal control problems are model-free and are classified as RL [8]. RL based control is designed for incrementally learning optimal control strategy that can maximize a reward function in the sequential decision-making setup. Compared to the traditional model predictive control, model-free RL does not require the prior system and can also work naturally in a stochastic system environment [9].

RL algorithms have also been applied in exoskeleton control in some latest literature. In [10], the policy iteration is combined with dynamic movement primitives to eliminate the effects of uncertainties in joint space for a two-leg walking exoskeleton robot. Hamaya et al. [11] utilized the electromyography signals to measure the user's intended movements, and adopt RL to derive control policy for designing walking assistive strategies.

## 2 Problem Formulation

Our aim is to design an adaptive controller for a single-side lower-limb exoskeleton with 2 actuated degrees of freedom (as depicted in Figure 1(a), 1 at the hip flexion/extension and 1 at the knee flexion/extension) to provide appropriate walking assistance for people suffering from unilateral weakness or even hemiparesis. Such subjects usually suffer disabled walking ability in one leg and remain normal for the other leg. However, symmetry is expected in moving pattern of two legs during locomotion. Therefore, we propose to model the exoskeleton system as a leader-follower system, where the unaffected leg is treated as leader and the affected side with exoskeleton is treated as the follower. For the control of the exoskeleton of the affected leg, we propose to use the RL based adaptive PD controller framework for accurately control.

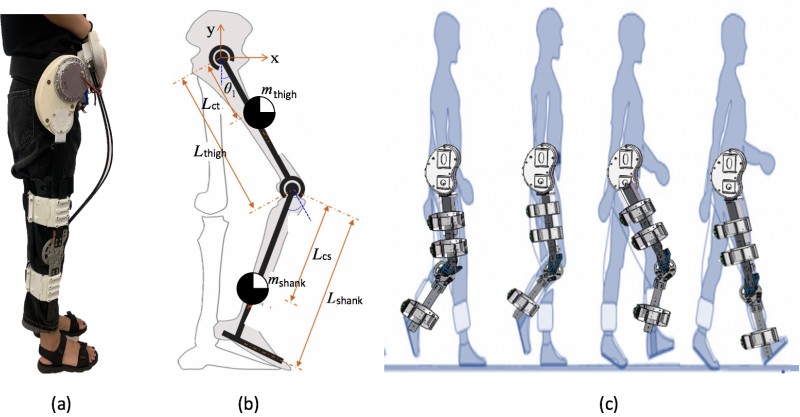

Figure 1: The exoskeleton-human system: (a) The prototype of our exoskeleton system. (b) Dynamic model of the exoskeleton-human system. (c) Leader-follower system: the affected side (with exoskeleton) is treated as follower while the unaffected side is treated as leader.

As show in Figure 1(b), the single-side exoskeleton system can be modeled as a two-joint robot and the dynamics can be formulated as a general nonlinear mechanical system (i.e., Euler-Lagrange system) as follows:

$$\tau = \frac{d}{dt}\frac{\partial L}{\partial \dot{\theta}} - \frac{\partial L}{\partial \theta} \tag{1}$$

where $\tau$ denotes the generalized force or torque, $L$ denotes Lagrange function which equals to difference of the system kinetic energy with potential energy (i.e., $L = K - P$), $\theta = (\theta_{hip}, \theta_{knee})^T \in \mathbb{R}^2$ denotes generalized coordinates with value equals to the joints' angles of the exoskeleton while $\theta_{hip}$ and $\theta_{knee}$ represent the angle of hip and the knee respectively, and $\dot{\theta} = (\dot{\theta}_{hip}, \dot{\theta}_{knee})^T$ is the corresponding angular velocity.

The dynamics of the unaffected side (the leg without exoskeleton, as shown in Figure 1(c)) which serves as the reference of motion trajectory is expressed by:

$$\dot{\theta}_{ref} = f(\theta_{ref}) \tag{2}$$

Then the walking assistance control problem is transformed into an optimal control problem, which aims at designing a distributed controller to guarantee that the motion trajectory of the exoskeleton being able to track the unaffected leg's motion of last gait cycle. We consider to present the RL task in a Markov Decision Processes (MDP) with continuous action space, which can be defined as a tuple of states, actions, transition probabilities, policy, rewards and discounting factor $\{\mathcal{S}, \mathcal{A}, \mathcal{T}, \mathcal{P}, \mathcal{R}, \gamma\}$. Here, the agent receives a state observation $s = \left\{\theta, \dot{\theta}, \ddot{\theta}, \left\{\theta_{dsr}, \dot{\theta}_{dsr}\right\}_{\mathbf{T}}\right\}$, which are defined as the actual angle, angular velocity, angular acceleration measured with encoders, and the $\left\{\theta_{dsr}, \dot{\theta}_{dsr}\right\}_{\mathbf{T}}$ is the desired trajectory of a time period obtained with inertial measurement units (IMUs) tightly attached on the unaffected leg. All the states responds with an action $\tau = \left\{\begin{array}{c}\tau_{hip}\\ \tau_{knee}\end{array}\right\} \in \mathcal{A}$ which represents the possible control torque of motor at each joint. The transition model $\mathcal{T}$ is given by dynamics of the exoskeleton-human system. The return reward $R$ from a state is defined as function indicating tracking errors, and a discounting factor $\gamma \in [0, 1]$ is introduced to the reward learnt by the critic network.

## 3 Experiment Setup

For learning-based controllers on robots, especially in our application scenario of exoskeletons, unexpected ill performance may leads to severe results. Therefore, we build a simulation platform based on the frame of pyglet library to validate the proposed controller. As the action space is continuous in our application, policy gradient based algorithms like Proximal Policy Optimization (PPO), Deep Deterministic Policy Gradient (DDPG) and Twin Delayed DDPG (TD3) are considered to model the controller. Through preliminary experiments, we develop a TD3 based controller that can make the follower tracking the leader leg.

### 3.1 Simulation Platform

The typical observations for exoskeleton controllers are sensory data, e.g. electromyogram (EMG) signals [12], under-footplate pressure [13]. In light of this, we utilize the angle, angular velocity and acceleration of each joint that can be obtained by IMU and encoders as observations.

For the dynamics of the developed exoskeleton-human system, equation 1 can be further written in matrix form as follows [14], considering the human-exoskeleton interaction and disturbances:

$$M(q)\ddot{q} + C(q, \dot{q})\dot{q} + G(q) = \tau_{actuator} + \tau_{human} + \tau_{dist} \tag{3}$$

Here, in simulation case, the given dynamic system is used to produce system data for the adaptive strategy. The system matrices are given as follows:

$$M(q) = \begin{bmatrix} m_t l_{ct}^2 + m_s(l_t^2 + l_{cs}^2 + 2l_t l_{cs}c(\theta_{knee})) + I_t + I_s & m_s(l_{cs}^2 + l_t l_{cs}c(\theta_{knee})) + I_s \\ m_s(l_{cs}^2 + l_t l_{cs}c(\theta_{knee})) + I_s & m_s l_{cs}^2 + I_s \end{bmatrix} \tag{4}$$

$$C(q, \dot{q}) = \begin{bmatrix} -2m_s l_t l_{cs} s(\theta_{knee})\dot{\theta}_{knee} & -m_s l_t l_{cs} s(\theta_{knee})\dot{\theta}_{knee} \\ m_s l_t l_{cs} s(\theta_{knee})\dot{\theta}_{hip} & 0 \end{bmatrix} \tag{5}$$

$$G(q) = \begin{bmatrix} m_t g l_{ct} s(\theta_{hip}) + m_s g l_{cs} c(\theta_{hip} - \theta_{knee}) \\ -m_s g l_{cs} s(\theta_{hip} - \theta_{knee}) \end{bmatrix} \tag{6}$$

where $c$ represents $cos$, $s$ represents $sin$, $m_t$ and $m_s$ denotes the mass of thigh and shank, $l_t$ and $l_s$ denotes the length of thigh and shank, $l_{ct}$ and $l_{cs}$ denotes the distance from the centre of mass of thigh to the adjacent hip joint and that of shank to the knee joint respectively. For each subject, a system identification process is taken for identifying these matrices by collecting data of the exoskeleton system and the details are not introduced here as we focus on the platform development and learning based control.

Considering the specifications of mechanical design and electric system (some related technical data is provided in Table 1), we further introduce some important constraints. First of all, as there is limit on maximum output torques, an action bound is adopted to restrict the distributed output torques. Also, the value of $\dot{\theta}$ of each joint is clipped to the maximum angular velocity requirements. And every time the limit is meet, a severe punishment of rewards -100 will be exerted. Finally, though we have mechanical hard stop in the design of exoskeleton to ensure safety, we still do not want the joint moves to its limit. If the actual angle $\theta$ exceeds the ROM (range of motion), the simulation platform will give a punishment of rewards -1000 and break out the current episode.

Table 1: Specification Parameters of the Exoskeleton System

| Significant System Technical Parameters | Exoskeleton Module | |
|---|---|---|
| | Hip | Knee |
| Transmission Ratio | 4.90 | 4.44 |
| Maximum Output Torque[a] (Nm) | 33.8 | 30.6 |
| Maximum Angular Velocity[a] (rad/s) | 7.12 | 7.86 |
| Range of Motion (deg) | [-10, 110] | [0, 135] |

[a]*Calculated based on technical parameters of the adopted electric motor and transmission ratio.*

It is worth noted that as the sampling frequency of motor is 2 kHz and the communication frequency of the CAN protocol is 1 MHz that are vastly higher than the sampling frequency of 50 Hz of the IMU sensors used to obtain the desired trajectory from the leader leg. To achieve smooth control, we set the frequency of the motor as 10 times of the sampled desired trajectory, thereby obtaining a sparse and delayed reward.

The workflow of the simulation platform is shown in Figure 2: (1) A simulated exoskeleton-human system is initialized with the loaded human information; (2) The controller output torques according to current policy and state observations; (3) The states and dynamic model of the exoskeleton-human system is updated as a result of the exerted torques under the system dynamics illustrated above; (4) The transitions are observed and utilized to determine the optimal policy. The process (2) to (4) is repeated during the walking process.

### 3.2 Reinforcement Learning based Adaptive Controller

In this exoskeleton continuously following control problem, we utilize PD controller as our baseline controller, because the bias of the whole system is highly affected by the position error and velocity error of the two joints. The principle of PD control is:

$$\tau = K_p(\theta_{goal} - \theta) + K_d(\dot{\theta}_{goal} - \dot{\theta}), \tag{7}$$

where $\theta_{goal}$ is the predefined or real time measured goal position of the two joints. The PD controller can perform well when the goal trajectory is within a smooth derivative curve. For example, based on empirical data, we can model the goal position as:

$$\theta_{leader} = \begin{bmatrix} Norm_{min-max}(0.5cos(t) + 0.2sin(3t)) * 2.3 + 2.6 \\ Norm_{min-max}(0.3cos(3t) - 0.5sin(2t)) * 1.6. \end{bmatrix} \tag{8}$$

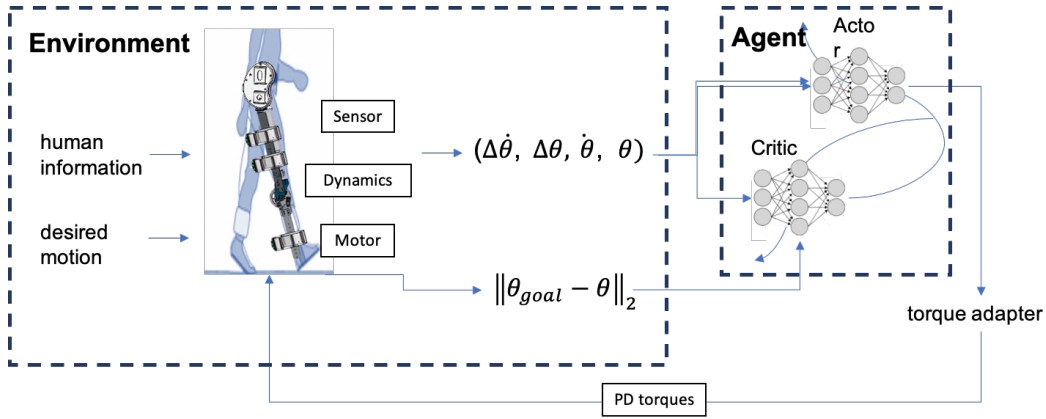

Figure 2: Workflow of the simulation: the left dash box denotes the simulation environment which responds action with state transitions, while the right dash box denotes the RL-based assistance controller that takes in system observations and outputs torques.

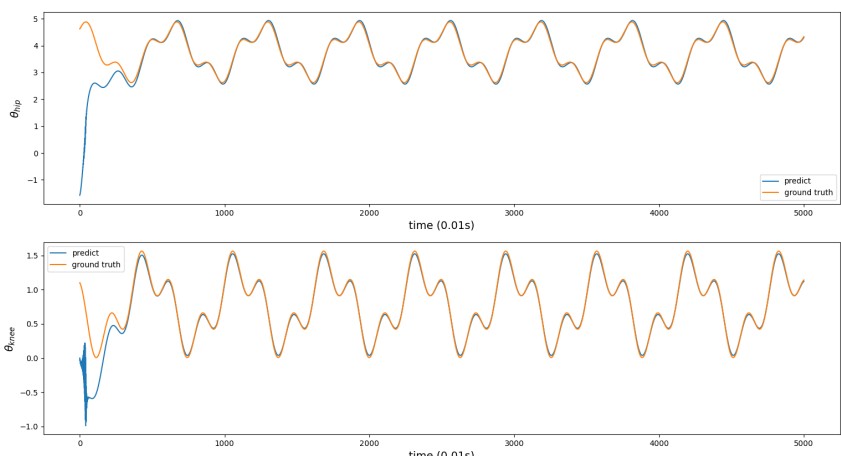

Figure 3: Tracking Performance of simulated trajectory

Then different parameters of the PD controller were tried, and finally the $K_p = diag(100, 100)$, $K_d = diag(50, 50)$ performs well and the trajectory is shown as in Figure 3. However, in real life, neither the trajectory nor the velocity of the joint is with in regional monotony. Thus, we utilize the RL technique to dynamic adaptive the amplitude of the $\tau$ calculated based on equation 4a,of which the optimal value is highly affected by the feedback real status of the joints, i.e., states in RL.

Based on the task requirement, our RL system should have two dimensional output to dynamic adjust the output of the PD controller. The torque provided by the exoskeleton motor should be within a safety range. Thus, we apply our action bound as $[0, 2]$, and then the real torque applied to the environment is: $\tau_{real} = a * \tau$. Because of the measurement sampling rate limitation, cubic interpolation to the measured angles of the joints is utilized to produce plentiful of torque baseline produced by PD controller. For the rewards, we utilize the sparse reward, i.e., 10 actions with one reward. We then try to evaluate the performance of different RL algorithms.

## 4   Experiments and Results

In this section, we will illustrate our trial on how RL improve the tracking performance of our exoskeleton-human system. In the simulation environment, the torque is transformed to the next state of the leg via Dynamics. The reference trajectory of the leader leg is transferred from the motion trajectory of the unaffected leg in the before-mentioned leader-follower system with a time interval. To better simulate the real environment, we use the motion trajectory obtained from the

IMU sensors attached as desired trajectory, which is also feed into the RL based controller as part of the observations. And the information from encoder and motor are calculated with the dynamics and kinematics in the simulation platform. Based on the developed simulation platform, we conducted a series of experiments with various parameter settings and our proposed RL based controller outperforms the classical PD controllers.

Different parameters of PD controller are applied first. However, the angle gap between the leader and the follower can not be effectively suppressed, as shown in Figure 4a. We analyze this phenomenon is caused by the non-uniform velocity during human walking patterns. To improve the tracking accuracy of the follower, we utilized RL to adaptive shift the amplitude of the torque generated by PD controller.

The tracking performance of the PPO adaptive control is shown as in Figure 4b. We expend the feature size by using the 50 frames of the states as the network input. The Temporal Convolutional Network (TCN) and the output layer is applied for the distribution and critic generation. TCN has four residual blocks with 16,32, 64,128, and the kernel size 3 leads to the receptive field of view 61, which is sufficient enough to extract the whole input features. The Beta distribution is utilized to model the adjustment coefficient of the PD control generated torque. However, the tracking error has not been significantly reduced compared to original PD control.

We then consider the TD3 algorithm as the torque adapter. The target policy smoothing characteristic will theoretically made the implement torque more uniform, which is safer for the motor of the exoskeleton. The angle bias between the leader and follower is utilized as the reward for each step, while the interpolated angle trajectories with 10 times higher sampling rate is utilized as the desired trajectory. Here we made a small trick by allowing the adapter to change the sign of the PD control generated torque, i.e., the action coefficient bound generated by the RL is $[-0.1, 2]$. The angles of the leader and follower is shown as in Figure 4c. To make the rewards be consistent with the measured angles of the leader, we then apply spare rewards to the TD3 algorithm, i.e., 10 actions with 1 reward, and the results is shown in Figure 4d. It is obvious that sparse rewards produce high tracking accuracy, especially for the knee angle, shown as in Figure 5.

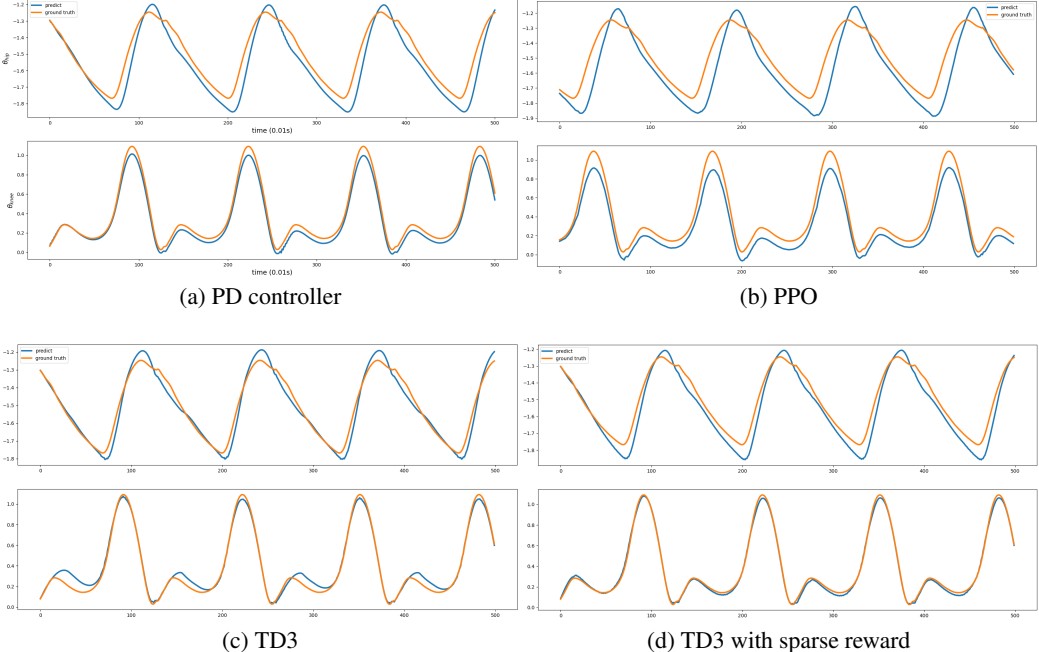

Figure 4: Tracking trajectories of collected trajectory by different algorithms.

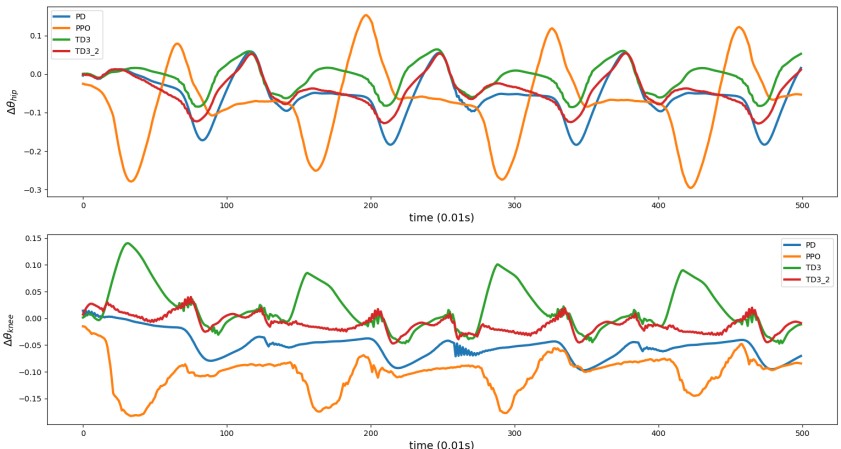

Figure 5: Visualization of the tracking error for each algorithms.

## 5 Conclusion and Future Work

In this project, we applied the RL techniques in walking assistance control of a single-side lower-limb exoskeleton system. As the exoskeleton system is designed by ourselves, there is no existed simulation environment suitable for our problem, we build a simulation platform for our exoskeleton-human system based on the Pyglet library. Then we design the RL based adaptive PD controller taking the pre-collected sensor sensor readings from IMU and encoder as observations to give desired angles and velocities for both joint of the exoskeleton-human system. According to the application requirements of hemiparesis walking assistance, the ultimate aim is to track the desired trajectory obtained from the healthy leg which serves as leader in the modeled system in real time. With the TD3 based algorithms, our controller outperforms the classical PD controller in terms of tracking error.

There is few literature using RL techniques in control of exoskeletons, most of which focus on conducting system identification process and intention based control. To our knowledge, our idea and implementations is a first try to apply such techniques in directly modeling the controller to track the time-variant motion trajectory. However, we only compare the proposed controller with classical PD controller and there is still a notable tracking error. And we found that the performance of the RL based controller really depends on fine-tuning of parameters through experiments, further fine-tuning may provide with better performance. Meanwhile, including more empirical knowledge should be considered, like applying RL techniques to further improve an oscillator-based controller.

In the future, we will also validate the proposed RL-based walking assistance control strategy through experiments on our real exoskeleton platform. Since the environment could be more complex which is unpredictable, further modifications on our developed simulation platform is also needed.

## 6 Acknowledgements

Thanks Professor Bolei Zhou and all the TAs for their hard working.

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
