# OpenReview forum: "Reinforcement Learning Based Adaptive WalkingAssistance Control of a Lower Limb Exoskeleton"
_CUHK.edu.hk/2021/Course/IERG5350_

### Official Review · AnonReviewer3 · 2020-12-19
**Creative work, but there are some structure problems of the paper.**

**Rating:** 7
**Confidence:** 3

**Review:**

Generally, this work is hard and creative, it uses different RL methods to control walking assistance in a single-side lower-limb
exoskeleton system. And the paper tests some RL algorithms' performance and adapts some tricks to the optimization of the problem. The simulation performs pretty good. However, there are still some problems in the text and structure which are worth noting.

Pros: Rich model design details. The problem definition is clear, It is easy to understand the goal of the model, and I expect you could further improve the performance of the walking assistance controller in more complex environments.

Cons: There are some problems in the text and structure problems which can make reader confused. Here are some listed problems and advices:

     1. The part of 'Related Work' seems a little short, the relevant literature should be sufficient.
         Advice: Maybe you could enrich 'Related Work' part.
     2. The Normalization of charts. The figure 4  lacks a description of the ordinate, and the size is inconsistent.
         Advice: you could reshape the figure and add supplement information.
     3. The citation and grammar issues.
         Advice: Avoiding using quotation as subject or object.
         For example:
              (1). In [10],the policy iteration is combined with dynamic movement primitives...
              (2). [11] utilized the  electromyography signals to measure the user’s intended movements.

These are my comments on your report. Good luck.

---

### Official Review · AnonReviewer2 · 2020-12-19
**The work is good, satisfying the requirements of the project**

**Rating:** 7
**Confidence:** 3

**Review:**

This paper proposes to control the single-side lower limb exoskeleton with 2 actuated degrees of freedom based on reinforcement learning. The proposed algorithm aims to provide optimal controlling signals to the leader-follower system, which essentially reformulates this controlling problem as a motion trajectory tracking task.

Clarity:
This paper is well written and easy to follow.

Originality:
This is the first time I have seen RL is applied to the single-side lower limb exoskeleton controlling task. Maybe because this task is a little bit small.

Pros:

The problem formulation is reasonable and practically feasible.  The experiment design is systematic and sufficient to verify the superiority of the proposed method compared with the classic methods such as PD controller.

Cons:

It seems that the experiment is done on some periodic and relatively simple data (without noise) as shown in Fig.3 and Fig.4. Have you ever tested your algorithm on some noisy and practical data?  Since in the realistic scenarios, the user's habit may diverse and the data might not be such ideal especially in complex environments.

Some typos errors:

make the follower tracking (track) the leader leg

e.g. electromyogram (EMG) signals [12], (and) under-footplate pressure [13]

It is worth noted (noting)

---

### Official Review · AnonReviewer1 · 2020-12-20
**A good paper that contributes non-trivial advances over prior works in the field**

**Rating:** 7
**Confidence:** 4

**Review:**

Summary:
The authors proposed to address the limitations in existing approaches to developing walking assistance systems, such as the vulnerability to disturbances and the lack of adaptability, through a reinforcement-learning-based motion controller. In particular, they adopted a leader-follower model to capture the interactions between the exoskeleton system and the user, where the unaffected side is treated as the leader and the affected side augmented with exoskeleton is treated as the follower, and reformulated the walking assistance control problem into a motion trajectory tracking problem. The authors implemented a simulation platform based on the pyglet library to evaluate their RL-based controller. The resulting TD3-based controller achieved better tracking performance than traditional PD controllers.

Novelty:
This paper contributed non-trivial advances over prior works in the field, as the potential of RL-based walking assistance controllers has not yet been thoroughly studied.

Soundness:
I have carefully checked all details and did not find any significant technical flaws.

Impact:
This paper will impact a moderate number of researchers within the community.

Clarity:
While the paper is properly structured and reads well in general, the authors should consider putting more efforts into polishing the language, as the current version still contains quite a few grammatical and syntactic errors.

Evaluation:
The empirical validation looks good, though more results and analysis would significantly add support to the claims.

Pros:
1. The problem is well-defined, and the reformulation of walking assistance control into motion trajectory tracking makes sense.
2. This paper contributed non-trivial advances over prior works in the field.
3. The preliminary experimental results seem to be promising.

Cons:
1. The empirical results obtained in the simulation platform are not thorough enough to corroborate the claims made in the paper.
2. The current version still contains quite a few grammatical and syntactic errors.
3. The authors made a conceptual error on page 3. Rigorously speaking, DDPG and TD3 are not policy-gradient-based algorithms, but rather Q-learning-based.